# Reassessing the Impact of Coffee Consumption on Liver Disease: Insights from a Large-Scale Cohort Study with IPTW Adjustment

**DOI:** 10.3390/nu16132020

**Published:** 2024-06-26

**Authors:** Keungmo Yang, Young Chang, Soung Won Jeong, Jae Young Jang, Tom Ryu

**Affiliations:** 1Department of Internal Medicine, Division of Gastroenterology and Hepatology, College of Medicine, The Catholic University of Korea, Seoul 06591, Republic of Korea; 2Department of Internal Medicine, Institute for Digestive Research, Digestive Disease Center, Soonchunhyang University College of Medicine, Seoul 04401, Republic of Korea

**Keywords:** coffee, metabolic dysfunction-associated steatotic liver disease, survival, inverse probability treatment weighting

## Abstract

Coffee consumption is globally widespread and has become a lifestyle habit. This study investigated coffee consumption and liver-related survival in a large cohort of 455,870 individuals with UK biobank, categorized into without steatosis, metabolic dysfunction-associated steatotic liver disease (MASLD), and MASLD and increased alcohol intake (MetALD). Inverse probability of treatment weighting (IPTW) adjusted for confounding variables was used, followed by Kaplan–Meier analysis. Moderate coffee consumption (1–2 cups per day) was associated with lower all-cause mortality across the entire cohort, without the steatosis, MASLD (*p* < 0.0001), and MetALD cohorts (*p* = 0.0047 for pre-IPTW, *p* = 0.027 for post-IPTW). Before IPTW adjustment, consuming one or more cups of coffee per day appeared to significantly reduce liver-related mortality in the overall (*p* = 0.015) and MASLD cohorts (*p* = 0.011). However, post-IPTW application, no significant differences in liver-related mortality were observed between the coffee intake groups (*p* = 0.778, 0.319, 0.564, 0.238 for each group). While increased coffee consumption initially seemed to reduce liver-related mortality, after IPTW adjustment, only all-cause mortality significantly decreased (*p* < 0.0001 and *p* = 0.027). These findings suggest that previous studies might have overestimated the favorable effect of coffee intake on chronic liver disease due to confounding factors.

## 1. Introduction

Metabolic dysfunction-associated steatotic liver disease (MASLD) and MASLD and increased alcohol intake (MetALD) are emerging terms that describe liver diseases primarily driven by metabolic dysfunction [1]. These conditions encompass a spectrum of liver abnormalities ranging from simple steatosis to nonalcoholic steatohepatitis (NASH), which can progress to cirrhosis and hepatocellular carcinoma [2]. MASLD is characterized by the accumulation of fat in the liver in the absence of significant alcohol consumption, while MetALD includes liver disease associated with metabolic dysfunction, which may also overlap with other liver conditions such as alcohol-associated liver disease (ALD) [1].

Recent studies have highlighted the significant public health burden posed by these metabolic liver diseases. The global prevalence of MASLD is estimated to be around 30%, with more than a 50% increase from the 1990s to the 2010s, and its variations are based on geographic region, ethnicity, and the diagnostic criteria used [3,4]. MetALD, although less well-defined in the literature, is increasingly implicating a requirement for the specific management of the disease and future prospective studies for better defining its natural history [5]. As understanding of MASLD and MetALD evolves, there is a growing emphasis on identifying modifiable lifestyle factors, such as diet and physical activity, that can mitigate the diseases progression [6].

Coffee is one of the most widely consumed beverages globally, with an estimated 2.25 billion cups consumed daily, with approximately 400 million of those being in the United States [7]. This ubiquitous drink has previously been extensively studied for its potential health benefits and risks. A study suggested that moderate coffee intake was significantly related with a 15% reduction in the risk of cardiovascular disease, while higher levels of coffee consumption showed an increased risk [8].

Previous studies have suggested that coffee consumption may have a protective effect against the progression of liver diseases. A meta-analysis has shown that coffee drinking is associated with favorable effects for liver cirrhosis, finding that daily coffee consumption of 2 cups significantly reduces the risk of liver cirrhosis [9]. Furthermore, coffee was shown to decrease the risk of non-alcoholic fatty liver disease and its development to liver fibrosis [10,11].

This study aims to explore the impact of coffee intake on all-cause mortality and liver-related mortality, specifically focusing on MASLD, MetALD, and their associated clinical characteristics, using the large cohort in the UK biobank dataset and inverse probability of treatment weighting (IPTW), which is a statistical strategy used to adjust the potential biases introduced by confounding variables.

## 2. Materials and Methods

### 2.1. Collection of Patient Information Utilizing UK Biobank Cohort

Information was obtained through questionnaires, physical measurements, and biological samples. This study utilized data from UK Biobank (Application ID: 117214), focusing on a large-scale cohort to investigate the relationship between coffee consumption, all-cause mortality, and liver-related mortality using Data-Field number 1498.

This study was conducted in accordance with ethical guidelines, with all participants providing informed consent. Ethical approval was granted by the North West Multi-centre Research Ethics Committee (MREC).

### 2.2. Exclusion Criteria

In this study, populations with alcoholic, viral, and autoimmune liver diseases were excluded to isolate the effects of metabolic dysfunction on liver disease outcomes. This exclusion means that the associations between coffee consumption and liver disease are not confounded by other liver disease etiologies, allowing for a clearer evaluation of the impact of coffee on MASLD and MetALD.

### 2.3. Classification of Patient Cohorts

To specifically analyze the impact of coffee consumption, among the whole of participants in the UK biobank cohort, we excluded individuals with missing clinical information the other etiologies. The remaining population was classified into three groups based on the amount of coffee intake: none, 1 or 2 cups per day, or more than 3 cups per day. Subgroups without steatotic liver disease, with MASLD, and with MetALD were also classified into three groups based on the amount of coffee intake: none, 1 or 2 cups per day, or more than 3 cups per day.

### 2.4. Inverse Probability of Treatment Weighting

IPTW was applied to adjust for potential confounding variables. IPTW created a weighted sample in which the distribution of baseline covariates is balanced across different levels of coffee consumption. This method facilitated approximating a randomized controlled trial by accounting for observed confounders. Variables used in the weighting process included age, sex, body mass index (BMI), smoking status, alcohol intake, physical activity, and comorbidities such as type 2 diabetes, dyslipidemia, and hypertension. IPTW was performed using R software (version 4.4.0) with ‘twang’ and ‘survey’ packages. The IPTW-adjusted Kaplan–Meier curves were then plotted, and the log-rank test was repeated on the weighted samples.

### 2.5. Kaplan-Meier Survival Analysis

Statistical Kaplan–Meier survival analysis was employed to assess all-cause and liver-related mortality among the study participants. The follow-up duration spanned a median of 10 years, beginning from the baseline assessment and spanning until the occurrence of death or the end of the study period. Censoring was addressed for participants who were lost to follow-up or who were still alive at the end of the study. These individuals were considered right-censored at their last known contact date. The Kaplan–Meier method allowed us to estimate survival functions and visualize survival probabilities over time. To compare the survival distributions between different coffee consumption groups, we used the log-rank test. This statistical approach enabled us to evaluate the impact of coffee consumption on survival while appropriately handling censored data.

### 2.6. Statistical Anlysis

Statistical analyses were conducted to examine the relationship between coffee consumption and mortality outcomes with or without steatotic liver diseases. Initially, descriptive statistics were used to summarize baseline characteristics of the study population, stratified by coffee consumption levels using standardized means difference (SMD).

For the primary analysis, Kaplan–Meier survival curves were generated to visualize differences in all-cause and liver-related mortality across different coffee consumption groups. Liver-related mortality for the cause of death was defined as acute liver failure, acute on chronic liver failure, decompensated liver cirrhosis, and hepatocellular carcinoma. The log-rank test was employed to assess the statistical significance of survival differences between groups. Details for Kaplan–Meier survival curves were addressed in the previous section.

All statistical analyses were performed using R software (version 4.4.0). A *p*-value of <0.05 was considered statistically significant for all tests.

## 3. Results

### 3.1. Baseline Characteristics of Study Population

Baseline characteristics of the participants were categorized based on their coffee intake and the presence or absence of steatotic liver disease. The data were analyzed both before and after IPTW to ensure comparability across groups and the differences in baseline characteristics were minimized, with SMDs close to zero (Table 1, Table 2, Table 3 and Table 4). Participants were divided into three groups according to their coffee intake: none, 1–2 cups/day, and more than 3 cups/day.

For the entire cohort, the proportion of males with higher coffee consumption was: 42.1% in the no coffee group to 59.7% in the more than 3 cups/day group. After IPTW, the sex distribution became more balanced across groups (from 50.1% to 50.2%). The mean age at recruitment was similar across all groups, both before and after IPTW, with minor differences (around 57 years).

The smoking status varied across groups before IPTW, with current smokers being more prevalent in the more than 3 cups/day group (11.3%) compared to the no coffee group (10.3%). This difference was minimized after IPTW. Previous smokers were more prevalent in the 1–2 cups/day group, and never smokers were less prevalent in the 1–2 cups/day group.

The body mass index (BMI) and waist circumference showed minor variations between groups. Participants with no coffee intake had slightly higher BMI and waist circumference values compared to those with a higher coffee intake before IPTW, but these differences were negligible after IPTW.

Regarding metabolic health indicators, the prevalence of type 2 diabetes was slightly higher in the no coffee group (11.0%) before IPTW but was balanced across groups after IPTW (around 10.5%). The dyslipidemia and hypertension prevalence showed minimal differences across groups, with a slightly higher prominence in the higher coffee intake groups before IPTW, which was balanced after IPTW.

Liver function tests such as alanine aminotransferase (ALT) and gamma-glutamyl transferase (GGT) were slightly elevated in the no coffee group compared to the higher coffee intake groups before IPTW. These differences were also minimized after weighting. Platelet counts and albumin levels were consistent across all groups, showing no significant variation before or after IPTW (Table 1).

For participants without steatotic liver disease, the trends in baseline characteristics were similar to those for the overall cohort. Males constituted a higher proportion of the higher coffee intake groups, and the age distribution was consistent. Smoking status differences were also present but minimized after IPTW. The BMI and waist circumference values showed minor variations that balanced out after weighting. The prevalence of type 2 diabetes, dyslipidemia, and hypertension showed slight differences across groups but these were also balanced after IPTW. Liver function tests and other serological parameters remained consistent across groups (Table 2).

For participants with MASLD, similar patterns were observed. Males were more prevalent in the higher coffee intake groups, and the age distribution remained consistent. Smoking status differences were evident before IPTW but balanced after. The BMI and waist circumference differences were minimal. The prevalence of type 2 diabetes was higher in the no coffee group but balanced after IPTW. The prevalence of Dyslipidemia and hypertension showed minor differences, and liver function markers showed slight variations that were minimized after weighting (Table 3).

For the patients with MetALD, males were also prevalent in the higher coffee intake group and the age distribution also remained consistent. Differences in smoking status were balanced after IPTW adjustment. The BMI and waist circumference were negligible and differences in the presence of type 2 diabetes, dyslipidemia, and hypertension were also minimized after IPTW. Liver function markers and the other serological markers showed minimal variations and they were balanced after IPTW (Table 4).

Overall, major differences and minor variations in baseline characteristics were effectively balanced and minimized after IPTW in the entire cohort, and the cohorts without steatotic liver disease, with MASLD, and with MetALD.

### 3.2. Overall Survival in the Entire Cohort

Before IPTW, overall survival (OS) analysis showed that consuming less than 2 cups of coffee per day was associated with better survival (*p* < 0.0001). In contrast, liver-related survival was significantly better for individuals consuming more than 1 cup of coffee per day (*p* = 0.015) (Figure 1A,B). After applying IPTW, a moderate coffee intake of 1 or 2 cups per day was associated with a lower all-cause mortality rate (*p* < 0.0001), but there were no significant difference in survival rates between the different coffee intake groups (*p* = 0.778) (Figure 1C,D). These findings indicate that, after IPTW, the significance of coffee intake concerning liver-related survival diminishes in the entire cohort.

### 3.3. Coffee Intake and Mortality in the Cohort without Steatotic Liver Disease

We investigated the OS of a cohort without steatotic liver disease through subgroup analysis based on coffee intake. Kaplan–Meier survival analysis revealed that consuming less than 2 cups of coffee per day was associated with better OS for all-cause mortality before IPTW (*p* < 0.0001) (Figure 2A). While moderate coffee consumption (1 or 2 cups per day) tended to show better liver-related survival, the difference was not significant (*p* = 0.050) (Figure 2B). After IPTW, a daily intake of 1 or 2 cups of coffee was associated with significantly better all-cause mortality survival (*p* < 0.0001), whereas liver-related survival did not show significant differences (*p* = 0.319). These results suggest that, within a cohort without steatosis, coffee intake does not significantly alter survival rates when IPTW is applied.

### 3.4. Coffee Intake and Mortality in the MASLD Cohort

The third cohort consists of individuals with MASLD, representing steatotic liver disease related to metabolic disorders. Before IPTW, consuming more than 3 cups of coffee per day was associated with significantly worse all-cause mortality (*p* < 0.0001), whereas liver-related survival was better for those consuming 1 or 2 cups and more than 3 cups of coffee per day compared to non-coffee drinkers (*p* = 0.011) (Figure 3A,B). After IPTW, moderate coffee consumption (1 or 2 cups per day) was associated with better all-cause mortality rates (*p* < 0.0001), but liver-related survival showed no significant differences across coffee intake groups (*p* = 0.564) (Figure 3C,D). These findings suggest that, while moderate coffee consumption may improve all-cause mortality, it does not significantly affect liver-related mortality in the MASLD cohort.

### 3.5. Coffee Intake and Mortality in the MetALD Cohort

The final cohort examined is the MetALD population, characterized by MASLD with increased alcohol intake. Before IPTW, the OS for all-cause mortality was significantly higher in individuals consuming 1 or more cups of coffee per day compared to non-coffee drinkers (*p* = 0.0047), whereas the liver-related mortality did not significantly differ between groups with varying coffee intake levels (*p* = 0.440) (Figure 4A,B). After IPTW, a daily intake of 1 or 2 cups of coffee was associated with better all-cause mortality OS (*p* = 0.027), and there were no significant differences in survival rates between non-drinkers, those consuming 1 or 2 cups per day, and those consuming more than 3 cups of coffee per day (*p* = 0.238) in the MetALD cohort (Figure 4C,D). Collectively, these data suggest that moderate coffee consumption significantly enhances all-cause mortality survival but does not affect liver-related mortality in the MetALD population.

## 4. Discussion

The impact of coffee consumption on all-cause and liver-related mortality among various populations, specifically those with and without steatotic liver disease, MASLD, and MetALD, provides crucial insights into lifestyle modifications that could potentially improve health outcomes. This study, utilizing data from UK Biobank and applying inverse probability of treatment weighting (IPTW), aimed to address potential confounders and present more accurate associations.

In the overall cohort, before IPTW, consuming less than 2 cups of coffee per day was associated with better overall survival, while the liver-related survival was significantly better for individuals consuming more than 1 cup per day (*p* = 0.015). Post-IPTW, moderate coffee consumption (1–2 cups per day) was linked to lower all-cause mortality, though there were no significant differences in liver-related survival between the different coffee intake groups. These findings suggest that a moderate coffee intake may generally benefit overall survival but has a less clear impact on liver-related mortality after adjusting for confounders.

In the MASLD cohort, before IPTW, consuming more than 3 cups of coffee per day was associated with worse overall survival. Nevertheless, liver-related survival was likely to be better for those consuming more than 1 cup of coffee per day compared to non-coffee drinkers. After IPTW, moderate coffee consumption (1–2 cups per day) was still associated with better overall survival, while liver-related survival differences were not significant (*p* = 0.564). This suggests that, while moderate coffee intake improves the overall survival in MASLD patients, its impact on liver-related survival is less clear when adjusting for confounders.

Several mechanisms might explain the beneficial effects of coffee consumption on overall and liver-related mortality. Coffee contains numerous bioactive compounds, including caffeine, chlorogenic acids, and diterpenes, which have been shown to have anti-inflammatory, antioxidant, and antifibrotic properties [12,13,14].

Coffee is a significant source of dietary antioxidants, which can reduce oxidative stress, a key factor in liver disease progression and other metabolic disorders. Antioxidants help neutralize free radicals, reducing cellular damage and inflammation, which are critical in the pathogenesis of liver diseases such as MASLD and MetALD [15].

Meanwhile, chronic inflammation plays a crucial role in the progression of liver diseases. Coffee’s anti-inflammatory effects are partly due to its polyphenols and other bioactive compounds, which inhibit inflammatory pathways [16]. These anti-inflammatory properties could help reduce liver damage and slow disease progression. Also, coffee has been shown to affect liver enzymes, particularly ALT and GGT, which are markers of liver injury. Regular coffee consumption has been associated with lower levels of these enzymes, suggesting a protective effect against liver damage [17]. This modulation of liver enzymes may be one of the mechanisms by which coffee reduces the risk of liver disease progression.

Insulin resistance is another significant factor in the development and progression of MASLD. Coffee consumption has been linked to improved insulin sensitivity and a reduced risk of type 2 diabetes, which can indirectly benefit liver health by reducing the metabolic burden on the liver [18].

Finally, fibrosis is a critical step in the progression of liver disease to cirrhosis. Previous studies have suggested that coffee consumption can reduce liver fibrosis in patients with chronic liver diseases, including hepatitis C [19,20]. The anti-fibrotic effects may be mediated through coffee’s ability to reduce inflammation and oxidative stress, which are key drivers of fibrosis.

The findings of this study have significant public health implications, particularly given the high prevalence of MASLD and the rising incidence of MetALD. Lifestyle modifications, including dietary changes, are crucial in managing these conditions. The observed benefits of moderate coffee consumption suggest that incorporating coffee into the diet could be a simple, cost-effective strategy to improve overall survival.

Paradoxically, but crucially, this study challenges the previously well-established cellular, molecular, and clinical evidence supporting the beneficial effects of coffee on liver disease progression. By leveraging a large cohort and employing IPTW to adjust for confounding variables, our findings suggest that coffee consumption might not significantly impact liver-related mortality or the progression of liver disease as previously thought.

Our study, utilizing a large-scale cohort from UK Biobank, presents a more nuanced perspective. By applying IPTW, we adjusted for numerous potential confounders including age, sex, BMI, smoking status, alcohol intake, physical activity, and comorbidities such as type 2 diabetes, dyslipidemia, and hypertension. This methodological rigor helps in approximating the conditions of a randomized controlled trial, thereby providing a more accurate assessment of coffee’s effects.

The results indicate that, after IPTW adjustment, coffee consumption does not significantly impact liver-related mortality. This finding contrasts with earlier studies that did not account for such a comprehensive range of confounding factors. For instance, while pre-IPTW analysis showed a significant association between coffee intake and reduced liver-related mortality, this association dissipated after adjusting for confounders.

One possible interpretation is that the previously observed benefits of coffee may have been confounded by lifestyle and demographic factors. Individuals who consume coffee may also engage in other health-promoting behaviors or have different socio-economic statuses, which could contribute to their overall better health outcomes. By controlling for these variables, our study suggests that the direct impact of coffee on liver-related outcomes may be less substantial than previously thought. Moreover, our findings showed that moderate coffee consumption was associated with improved overall survival in the entire cohort and various subpopulations (without steatotic liver disease, MASLD, and MetALD). However, the lack of a significant impact on liver-related mortality post-IPTW indicates that coffee’s beneficial effects might be more generalized rather than specific to liver disease.

The present study has several important strengths. First, the large sample size from UK Biobank cohort provides substantial statistical power and enhances the utilization of our findings. Second, the comprehensive adjustment for confounders using IPTW allows for a more accurate estimation of the associations between coffee consumption and mortality outcomes by reducing the potential for bias from observed confounding variables. Finally, the use of a well-characterized cohort with detailed information on lifestyle, clinical, and demographic factors enables a thorough analysis of the impact of coffee consumption on both all-cause and liver-related mortality. These methodological strengths contribute to the validity of our study results.

Comparing our findings with previous studies, we observed that, while prior research often suggests a protective effect of coffee on liver-related mortality and disease progression [11], our study found that coffee consumption was associated with lower all-cause mortality but did not significantly impact liver-related mortality after IPTW adjustment. These differences could be due to methodological variations, such as the sample size, study design, and use of IPTW for confounder adjustment, which might reveal residual confounding effects in earlier studies. Additionally, population differences, including demographics and coffee consumption habits, as well as variations in the definition and measurement of coffee intake might contribute to these differences. These factors emphasize the complexity of studying coffee’s health effects and highlight the need for further research to elucidate the mechanisms underlying these associations.

Although moderate coffee consumption was associated with lower all-cause mortality, no significant differences in liver-related mortality were observed post-IPTW. This difference might be due to our comprehensive adjustment for confounders, suggesting that the observed benefits on liver-related mortality in previous studies might have been influenced by uncontrolled confounding factors. The mechanisms through which coffee affects overall health, such as its antioxidant and anti-inflammatory properties, may contribute more broadly to a reduction in general mortality rather than specifically to liver-related outcomes.

The health benefits of coffee are well-documented, including its antioxidant properties, anti-inflammatory effects, and improvement of insulin sensitivity, which contribute to overall health and reduce the risk of various chronic diseases, thereby lowering all-cause mortality. However, the lack of a significant impact on liver-related mortality post-IPTW suggests that these mechanisms may be less effective for liver-specific outcomes. While antioxidants and anti-inflammatory compounds theoretically protect against liver inflammation and fibrosis, the progression of liver disease involves complex interactions of various factors that might diminish these effects. The significant associations observed before IPTW adjustment might be affected by healthier behaviors and demographic factors common among coffee drinkers, which are not entirely related to coffee’s biological effects. In summary, while coffee consumption has clear health benefits due to its bioactive compounds, their direct impact on liver-related mortality may be limited by the multifactorial nature of liver disease. This emphasized the importance of further research to understand the specific pathways through which coffee influences liver health and to identify other modifiable factors that may more directly impact liver-related outcomes.

While our study has several strengths, including the large sample size and robust statistical methodology, it also has limitations. The observational nature of this study means that causality cannot be definitively established. Self-reported data on coffee consumption may introduce recall bias and inaccuracies, and this study did not differentiate between types of coffee or preparation methods, which could influence health effects.

Especially, we did not differentiate between types of coffee or their preparation methods. Different types of coffee, such as American coffee, espresso, arabica, robusta, caffeinated, and decaffeinated (and the decaffeination method), contain varying levels of bioactive compounds that can influence health outcomes differently. Preparation methods, including the addition of milk, sugar, or other additives, can also modify the health effects of coffee. This lack of differentiation may limit the utilization of our findings, as the specific health benefits observed could vary based on the type and preparation of the coffee consumed.

Furthermore, although this study utilized a large, diverse cohort from UK Biobank, the findings may not be generalizable to other populations or settings. Differences in genetic, environmental, and lifestyle factors across populations could influence the observed associations.

While IPTW is an effective method for adjusting for confounders in observational studies, it has several limitations. One significant limitation is the potential for residual confounding. This can occur if there are unmeasured confounders that the IPTW method cannot fully adjust for. For instance, lifestyle factors or genetic predispositions not accounted for in the model could influence both coffee consumption and liver-related outcomes, leading to biased results. Additionally, the effectiveness of IPTW relies heavily on the correct specification of the logistic regression model used to calculate the weights. Misspecification of this model could result in biased estimates and reduce the validity of the findings. Despite these limitations, IPTW would be a valuable method for enhancing the rigor of observational studies.

Future research should aim to corroborate these findings through randomized controlled trials to eliminate residual confounding. Studies should also explore the impact of different types of coffee and preparation methods to provide more detailed recommendations. Investigating the biological mechanisms behind coffee’s effects on liver health remains crucial to understanding its potential benefits.

## 5. Conclusions

In summary, our study is the largest cohort analysis to date examining the relationship between coffee consumption and liver-related outcomes, employing IPTW to account for confounding variables. The findings challenge the prevailing notion that coffee significantly reduces liver-related mortality, suggesting that its beneficial effects may be more related to overall survival rather than specific to liver health. This highlights the importance of rigorous methodological approaches in epidemiological research and calls for a re-evaluation of dietary recommendations concerning coffee consumption for liver disease patients.

## Figures and Tables

**Figure 1 nutrients-16-02020-f001:**
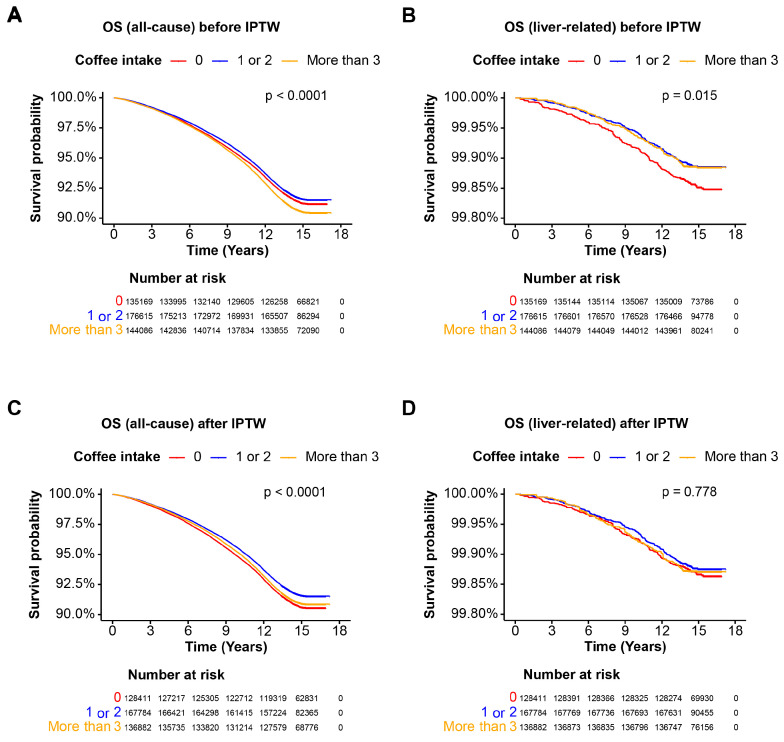
Overall survival of the entire cohort. Kaplan–Meier curves illustrating the overall survival (OS) probability over time for the entire cohort. Patients are categorized based on coffee intake: 0 cups/day, 1–2 cups/day, and more than 3 cups/day. The *x*-axis represents the time in years, and the *y*-axis represents the survival probability. The log-rank test *p*-value is indicated on the plot. (**A**) OS for all-cause mortality before IPTW; (**B**) OS for liver-related mortality before IPTW; (**C**) OS for all-cause mortality after IPTW; (**D**) OS for liver-related mortality after IPTW. Abbreviations: OS, overall survival.

**Figure 2 nutrients-16-02020-f002:**
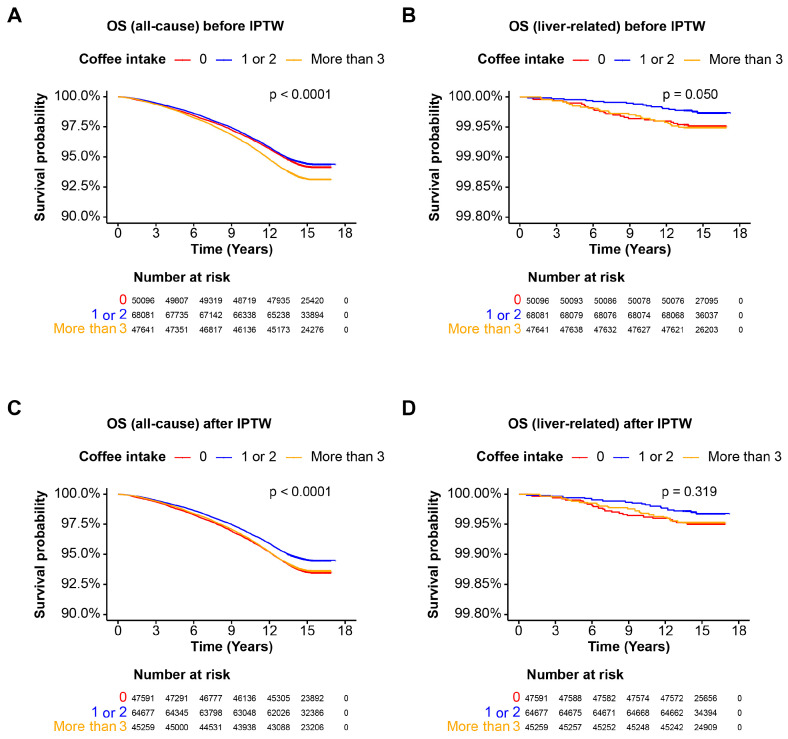
Overall survival of the cohort without steatosis. Kaplan–Meier curves illustrating the OS probability over time for the cohort without steatotic liver disease. Patients are categorized based on coffee intake: 0 cups/day, 1–2 cups/day, and more than 3 cups/day. The *x*-axis represents the time in years, and the *y*-axis represents the survival probability. The log-rank test *p*-value is indicated on the plot. (**A**) OS for all-cause mortality before IPTW; (**B**) OS for liver-related mortality before IPTW; (**C**) OS for all-cause mortality after IPTW; (**D**) OS for liver-related mortality after IPTW. Abbreviations: OS, overall survival.

**Figure 3 nutrients-16-02020-f003:**
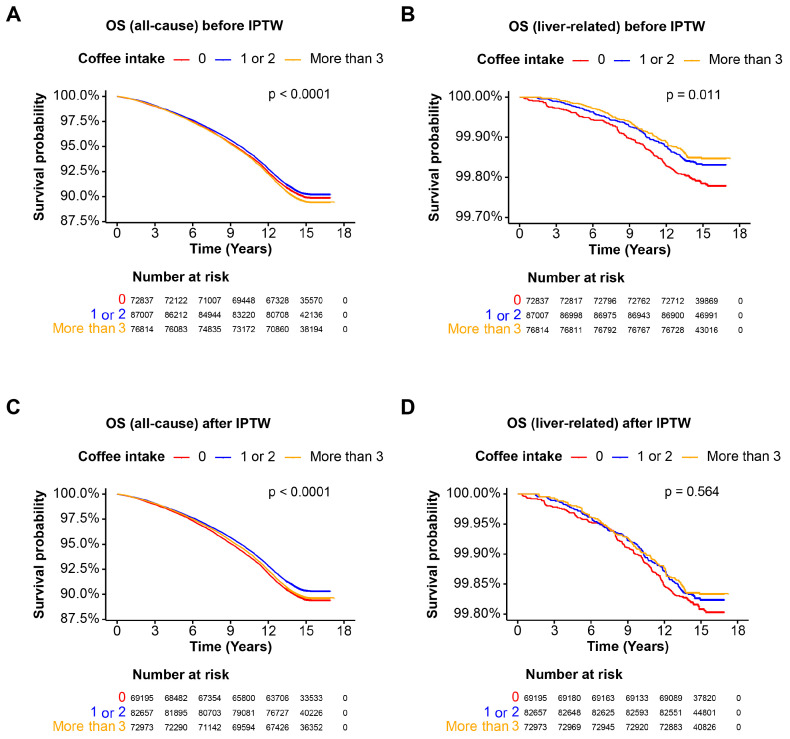
Overall survival of the MASLD cohort. Kaplan–Meier curves illustrating the OS probability over time for the MASLD cohort. Patients are categorized based on coffee intake: 0 cups/day, 1–2 cups/day, and more than 3 cups/day. The *x*-axis represents the time in years, and the *y*-axis represents the survival probability. The log-rank test *p*-value is indicated on the plot. (**A**) OS for all-cause mortality before IPTW; (**B**) OS for liver-related mortality before IPTW; (**C**) OS for all-cause mortality after IPTW; (**D**) OS for liver-related mortality after IPTW. Abbreviations: MASLD, metabolic dysfunction-associated steatotic liver disease; OS, overall survival.

**Figure 4 nutrients-16-02020-f004:**
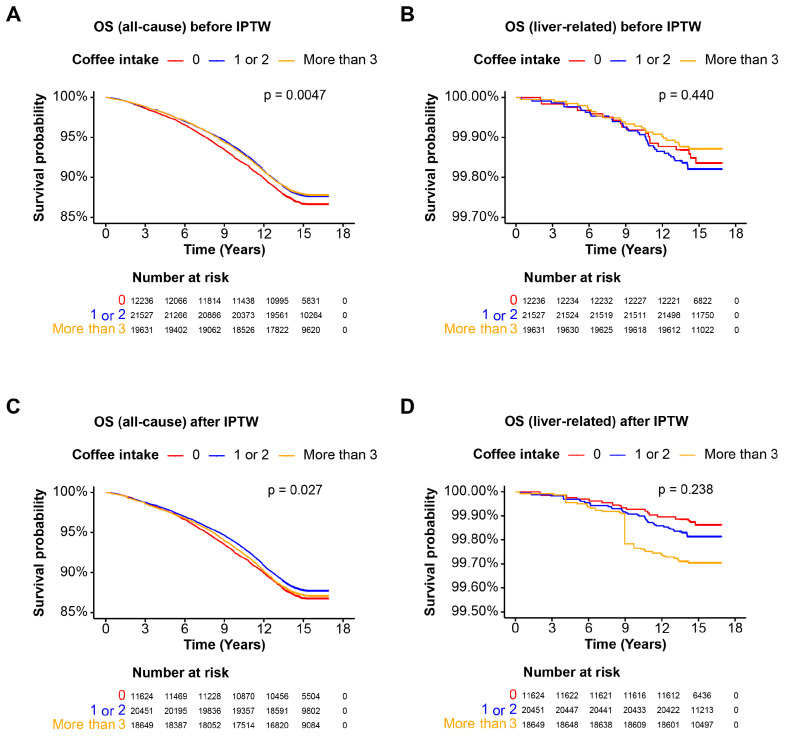
Overall survival of the MetALD cohort. Kaplan–Meier curves illustrating the OS probability over time for the MetALD (MASLD and increased alcohol intake) cohort. Patients are categorized based on coffee intake: 0 cups/day, 1–2 cups/day, and more than 3 cups/day. The *x*-axis represents the time in years, and the *y*-axis represents the survival probability. The log-rank test *p*-value is indicated on the plot. (**A**) OS for all-cause mortality before IPTW; (**B**) OS for liver-related mortality before IPTW; (**C**) OS for all-cause mortality after IPTW; (**D**) OS for liver-related mortality after IPTW. Abbreviations: MetALD, MASLD and increased alcohol intake; OS, overall survival.

**Table 1 nutrients-16-02020-t001:** Baseline clinical characteristics of the entire cohorts.

Clinical Characteristics	Before IPTW				After IPTW			
	Coffee Intake, Cups/Day			SMD	Coffee Intake, Cups/Day			SMD
	None, *n* = 135,169	1 or 2, *n* = 176,615	More than 3, *n* = 144,086		None, *n* = 128,410	1 or 2, *n* = 167,784	More than 3, *n* = 136,882	
**Sex (Male)**	56,913 (42.1%)	77,148 (43.7%)	71,266 (49.5%)	0.124	58,220 (45.3%)	75,596 (45.1%)	61,841 (45.2%)	0.001
**Age at recruitment**	55.71 ± 8.20	57.21 ± 8.06	56.53 ± 8.01	0.099	56.58 ± 8.09	56.53 ± 8.20	56.60 ± 8.01	0.006
**Smoking status**				0.175				0.002
Current	12,517 (9.3%)	13,862 (7.8%)	20,825 (14.5%)		13,367 (10.4%)	17,512 (10.4%)	14,253 (10.4%)	
Previous	42,599 (31.5%)	61,882 (35.0%)	52,121 (36.2%)		44,290 (34.5%)	57,659 (34.4%)	47,115 (34.4%)	
Never	79,932 (59.2%)	100,871 (57.1%)	71,140 (49.4%)		70,720 (55.1%)	92,614 (55.2%)	75,514 (55.2%)	
**Body mass index (kg/m^2^)**	27.42 ± 4.98	27.09 ± 4.65	27.77 ± 4.77	0.095	27.41 ± 4.90	27.42 ± 4.88	27.44 ± 4.65	0.004
**Waist circumference (cm)**	89.80 ± 13.65	89.32 ± 13.22	91.45 ± 13.50	0.105	90.18 ± 13.57	90.17 ± 13.54	90.21 ± 13.36	0.002
**Type 2 diabetes**	15,501 (11.5%)	16,891 (9.6%)	15,147 (10.5%)	0.042	13,420 (10.5%)	17,595 (10.5%)	14,333 (10.5%)	0.001
**Dyslipidemia**	37,979 (28.1%)	50,189 (28.4%)	41,802 (29.0%)	0.016	36,765 (28.6%)	47,851 (28.5%)	39,220 (28.7%)	0.002
**Hypertension**	57,056 (42.2%)	72,461 (41.0%)	59,335 (41.2%)	0.013	53,377 (41.6%)	69,540 (41.4%)	56,971 (41.6%)	0.002
**ALT (U/L)**	23.29 ± 14.00	23.16 ± 13.47	23.61 ± 13.92	0.022	23.40 ± 13.90	23.38 ± 13.54	23.45 ± 14.53	0.003
**GGT (U/L)**	37.84 ± 43.61	35.81 ± 38.09	36.00 ± 35.93	0.033	36.77 ± 37.48	36.56 ± 40.37	37.24 ± 47.28	0.011
**Platelet (10^9^/L)**	254.67 ± 61.08	252.01 ± 59.26	253.92 ± 59.60	0.059	253.36 ± 60.64	253.41 ± 60.18	253.41 ± 59.25	0.001
**Albumin (g/L)**	4.51 ± 0.27	4.53 ± 0.26	4.53 ± 0.26	0.030	4.52 ± 0.26	4.52 ± 0.26	4.52 ± 0.26	0.002

Data are described as mean ± standard deviation or *n* (%). ALT alanine transaminase; GGT, gamma-glutamyl transferase; SMD, standardized mean difference.

**Table 2 nutrients-16-02020-t002:** Baseline clinical characteristics of the participants without steatotic liver disease.

Clinical Characteristics	Before IPTW				After IPTW			
	Coffee Intake, Cups/Day			SMD	Coffee Intake, Cups/Day			SMD
	None, *n* = 50,096	1 or 2, *n* = 68,081	More than 3, *n* = 47,641		None, *n* = 47,591	1 or 2, *n* = 64,677	More than 3, *n* = 45,259	
**Sex (Male)**	10,797 (21.6%)	15,068 (22.1%)	12,049 (25.3%)	0.059	10,971 (23.1%)	14,770 (22.8%)	10,402 (23.0%)	0.003
**Age at recruitment**	53.99 ± 8.25	55.83 ± 8.23	55.36 ± 8.12	0.149	55.16 ± 8.23	55.11 ± 8.32	55.18 ± 8.15	0.005
**Smoking status**				0.200				0.002
Current	4101 (8.2%)	4665 (6.9%)	6899 (14.5%)		4542 (9.6%)	6162 (9.5%)	4311 (9.5%)	
Previous	48,402 (29.2%)	13,244 (26.5%)	20,333 (29.9%)		13,950 (29.3%)	18,895 (29.2%)	13,227 (29.2%)	
Never	32,630 (65.3%)	43,083 (63.3%)	25,917 (54.4%)		29,064 (61.1%)	39,620 (61.3%)	27,721 (61.3%)	
**Body mass index (kg/m^2^)**	23.42 ± 2.39	23.47 ± 2.30	23.72 ± 2.36	0.085	23.53 ± 2.38	23.53 ± 2.31	23.53 ± 2.36	0.002
**Waist circumference (cm)**	77.26 ± 7.13	77.51 ± 7.07	78.12 ± 7.10	0.081	77.62 ± 7.12	77.61 ± 7.07	77.63 ± 7.16	0.002
**Type 2 diabetes**	1569 (3.1%)	1800 (2.6%)	1461 (3.1%)	0.019	1398 (2.9%)	1883 (2.9%)	1333 (2.9%)	0.001
**Dyslipidemia**	7229 (14.4%)	10,893 (16.0%)	7612 (16.0%)	0.029	7410 (15.6%)	10,009 (15.5%)	7049 (15.6%)	0.002
**Hypertension**	12,195 (24.3%)	17,089 (25.1%)	11,776 (24.7%)	0.012	11,816 (24.8%)	15,977 (24.7%)	11,250 (24.9%)	0.002
**ALT (U/L)**	17.27 ± 7.57	17.59 ± 7.61	17.44 ± 7.95	0.027	17.48 ± 7.78	17.47 ± 7.43	17.52 ± 8.66	0.004
**GGT (U/L)**	21.77 ± 13.64	21.57 ± 12.69	21.48 ± 13.09	0.015	21.66 ± 12.60	21.59 ± 12.97	21.80 ± 16.58	0.010
**Platelet (10^9^/L)**	253.99 ± 59.53	252.97 ± 57.87	256.13 ± 59.28	0.036	254.21 ± 59.48	254.17 ± 58.95	254.19 ± 58.41	<0.001
**Albumin (g/L)**	4.51 ± 0.26	4.54 ± 0.26	4.54 ± 0.26	0.070	4.53 ± 0.26	4.53 ± 0.26	4.53 ± 0.26	<0.001

Data are described as mean ± standard deviation or *n* (%). ALT alanine transaminase; GGT, gamma-glutamyl transferase; SMD, standardized mean difference.

**Table 3 nutrients-16-02020-t003:** Baseline clinical characteristics of patients with metabolic dysfunction-associated steatotic liver disease (MASLD).

Clinical Characteristics	Before IPTW				After IPTW			
	Coffee Intake, Cups/Day			SMD	Coffee Intake, Cups/Day			SMD
	None, *n* = 72,873	1 or 2, *n* = 87,007	More than 3, *n* = 76,814		None, *n* = 69,195	1 or 2, *n* = 82,657	More than 3, *n* = 72,973	
**Sex (Male)**	37,568 (51.6%)	47,082 (54.1%)	44,923 (58.5%)	0.093	38,049 (55.0%)	45,277 (54.8%)	39,981 (54.8%)	0.003
**Age at recruitment**	56.45 ± 8.06	57.74 ± 7.93	56.74 ± 7.96	0.108	57.05 ± 7.98	56.99 ± 8.10	57.05 ± 7.91	0.005
**Smoking status**				0.164				0.002
Current	6736 (9.2%)	6863 (7.9%)	10,770 (14.0%)		7138 (10.3%)	8553 (10.3%)	7535 (10.3%)	
Previous	23,419 (32.2%)	30,838 (35.4%)	28,046 (36.5%)		24,141 (34.9%)	28,754 (34.8%)	25,400 (34.8%)	
Never	42,682 (58.6%)	49,306 (56.7%)	37,998 (49.5%)		37,916 (54.8%)	45,350 (54.9%)	40,039 (54.9%)	
**Body mass index (kg/m^2^)**	29.97 ± 4.68	29.62 ± 4.41	30.06 ± 4.50	0.066	29.87 ± 4.62	29.89 ± 4.63	29.89 ± 4.37	0.003
**Waist circumference (cm)**	97.21 ± 11.06	96.79 ± 10.70	98.22 ± 11.00	0.087	97.40 ± 11.02	97.41 ± 11.04	97.41 ± 10.77	0.001
**Type 2 diabetes**	12,484 (17.1%)	12,823 (14.7%)	11,628 (15.1%)	0.044	10,784 (15.6%)	12,940 (15.7%)	11,358 (15.6%)	0.002
**Dyslipidemia**	26,082 (35.8%)	31,135 (35.8%)	27,077 (35.3%)	0.008	24,708 (35.7%)	29,440 (35.6%)	26,020 (35.7%)	0.001
**Hypertension**	37,902 (52.0%)	43,942 (50.5%)	37,654 (49.0%)	0.04	35,011 (50.6%)	41,744 (50.5%)	36,899 (50.6%)	0.001
**ALT (U/L)**	26.56 ± 15.40	26.45 ± 14.85	26.47 ± 15.03	0.005	26.52 ± 15.16	26.53 ± 14.75	26.57 ± 15.97	0.002
**GGT (U/L)**	44.29 ± 46.16	42.31 ± 41.63	41.22 ± 38.47	0.048	42.82 ± 40.33	42.64 ± 42.86	43.10 ± 47.99	0.007
**Platelet (10^9^/L)**	254.40 ± 61.11	256.37 ± 62.40	252.85 ± 60.77	0.038	254.34 ± 61.69	254.43 ± 61.55	254.41 ± 60.11	0.001
**Albumin (g/L)**	4.50 ± 0.27	4.51 ± 0.26	4.52 ± 0.26	0.054	4.51 ± 0.27	4.51 ± 0.26	4.52 ± 0.26	0.002

Data are described as mean ± standard deviation or *n* (%). ALT alanine transaminase; GGT, gamma-glutamyl transferase; SMD, standardized mean difference.

**Table 4 nutrients-16-02020-t004:** Baseline clinical characteristics of patients with MetALD.

Clinical Characteristics	Before IPTW				After IPTW			
	Coffee Intake, Cups/Day			SMD	Coffee Intake, Cups/Day			SMD
	None, *n* = 12,236	1 or 2, *n* = 21,527	More than 3, *n* = 19,631		None, *n* = 11,624	1 or 2, *n* = 20,451	More than 3, *n* = 18,649	
**Sex (Male)**	8548 (69.9%)	14,998 (69.7%)	14,294 (72.8%)	0.046	8255 (71.0%)	14,478 (70.8%)	13,214 (70.9%)	0.003
**Age at recruitment**	58.31 ± 7.44	59.50 ± 7.21	58.55 ± 7.42	0.108	58.87 ± 7.28	58.85 ± 7.47	58.86 ± 7.35	0.002
**Smoking status**				0.103	57.05 ± 7.98			0.003
Current	1680 (13.7%)	2334 (10.8%)	3156 (16.1%)		1569 (13.5%)	2757 (13.5%)	2521 (13.5%)	
Previous	5936 (48.5%)	10,711 (49.8%)	9250 (47.1%)		5664 (48.7%)	9921 (48.5%)	9049 (48.5%)	
Never	4620 (37.8%)	8482 (39.4%)	7225 (36.8%)		4392 (37.8%)	7773 (38.0%)	7080 (38.0%)	
**Body mass index (kg/m^2^)**	28.54 ± 3.79	28.35 ± 3.60	28.64 ± 3.65	0.053	28.52 ± 3.75	28.51 ± 3.71	28.52 ± 3.60	0.002
**Waist circumference (cm)**	96.97 ± 10.09	96.44 ± 9.75	97.29 ± 9.84	0.057	96.92 ± 9.98	96.88 ± 9.93	96.91 ± 9.81	0.003
**Type 2 diabetes**	1461 (11.9%)	2268 (10.5%)	2058 (10.5%)	0.031	1271 (10.9%)	2230 (10.9%)	2050 (11.0%)	0.002
**Dyslipidemia**	4688 (38.3%)	8161 (37.9%)	7113 (36.2%)	0.027	4358 (37.5%)	7649 (37.4%)	7013 (37.6%)	0.003
**Hypertension**	6988 (57.1%)	11,430 (53.1%)	9905 (50.5%)	0.089	6169 (53.1%)	10,845 (53.0%)	9911 (53.1%)	0.002
**ALT (U/L)**	28.48 ± 16.69	27.52 ± 15.87	27.41 ± 15.70	0.044	27.90 ± 16.57	27.75 ± 15.97	28.11 ± 17.77	0.014
**GGT (U/L)**	65.09 ± 74.15	54.59 ± 56.78	50.78 ± 49.29	0.153	56.93 ± 55.06	56.08 ± 64.77	58.39 ± 103.83	0.029
**Platelet (10^9^/L)**	247.37 ± 58.86	245.66 ± 56.98	247.10 ± 57.46	0.020	246.42 ± 58.14	246.54 ± 57.46	246.19 ± 57.67	0.004
**Albumin (g/L)**	4.53 ± 0.26	4.54 ± 0.26	4.54 ± 0.26	0.041	4.54 ± 0.26	4.54 ± 0.26	4.54 ± 0.26	0.005

Data are described as mean ± standard deviation or *n* (%). ALT alanine transaminase; GGT, gamma-glutamyl transferase; SMD, standardized mean difference.

## Data Availability

Data supporting the findings of this study are available from the corresponding author upon reasonable request.

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
