# Peer review of "Reassessing the Impact of Coffee Consumption on Liver Disease: Insights from a Large-Scale Cohort Study with IPTW Adjustment"

_nutrients, 2024, doi:10.3390/nu16132020_

Round 1

Reviewer 1 Report

Comments and Suggestions for Authors

The title accurately reflects the study's purpose and methods. This is an interesting article to read. I have some comments to strengthened and provide more valuable insights into the relationship between coffee consumption and liver disease outcomes.

·      The abstract provides a clear summary of the study but lacks details on the sample size and specific statistical results. Including these would strengthen the abstract.

·      The introduction is comprehensive but somewhat lengthy. Consider condensing some background information to focus more on the study's rationale.

·      It is unclear why certain populations were excluded. Providing more detail on the exclusion criteria and their justification would enhance transparency. 

·      While IPTW is a robust method for adjusting confounders, the manuscript lacks a detailed explanation of how weights were calculated and applied. Including this would help readers understand the robustness of the adjustment.

·      Mentioning potential limitations of IPTW, such as residual confounding, would provide a more balanced view.

·      More details on the Kaplan-Meier survival analysis, such as the censoring mechanism and the follow-up duration, would be beneficial.

·      The discussion provides a good interpretation of the findings but could better differentiate between all-cause and liver-related mortality. It should more explicitly address why liver-related mortality was not significantly impacted post-IPTW.

·      There should be a clearer distinction between findings from this study and previous studies. Explicitly discussing why this study's results differ from past research would be beneficial.

·      The proposed mechanisms for coffee's health benefits are well-explained but could be better linked to the study's findings. Consider discussing how the lack of impact on liver-related mortality fits with these mechanisms.

·      The limitations section is brief. It should acknowledge other potential limitations, such as the observational design, potential measurement error in self-reported coffee consumption, and lack of differentiation between types of coffee.

·      Discussing the generalizability of the findings to different populations or settings would be useful.

·      Consider adding a strengths section to highlight the robust aspects of the study, such as the large sample size and comprehensive adjustment for confounders.

Comments on the Quality of English Language

Several typographical  errors are noted throughout the manuscript. for example: line 91 - "da", day.

Author Response

Comments 1: The title accurately reflects the study's purpose and methods. This is an interesting article to read. I have some comments to strengthened and provide more valuable insights into the relationship between coffee consumption and liver disease outcomes.

Response 1: Thank you for your positive feedback on the title and your interest in our study. We appreciate your constructive comments aimed at strengthening the manuscript and providing more valuable insights into the relationship between coffee consumption and liver disease outcomes. Below, we addressed each of your comments in detail and outline the corresponding revisions we plan to make to enhance the clarity and importance of our study.

Comments 2: The abstract provides a clear summary of the study but lacks details on the sample size and specific statistical results. Including these would strengthen the abstract.

Response 2: Thank you for your insightful feedback regarding the abstract. We appreciate the importance of providing a comprehensive overview of the study, including specific details on the sample size and key statistical results. Including these details would indeed enhance the clarity and completeness of the abstract, allowing readers to better understand the scope and significance of our findings. We added the details for abstract

Comments 3: The introduction is comprehensive but somewhat lengthy. Consider condensing some background information to focus more on the study's rationale.

Response 3: Thank you for your valuable feedback on our introduction. We agree that a more concise introduction would improve the readability and focus of the manuscript. We reviewed the introduction section and condense the background information to emphasize the study's rationale with shortening the contents. 

Comment 4: It is unclear why certain populations were excluded. Providing more detail on the exclusion criteria and their justification would enhance transparency. 

Response 4: Thank you for your comment. We agree that providing detailed explanations for the exclusion criteria and their justification is crucial for enhancing the transparency and clarity of our study. We added details in method section. This information would help readers understand the rationale behind our study design and the selection of the study population.

Comment 5: While IPTW is a robust method for adjusting confounders, the manuscript lacks a detailed explanation of how weights were calculated and applied. Including this would help readers understand the robustness of the adjustment.

Response 5: Thank you for your valuable feedback. We agree that providing a detailed explanation of how the weights were calculated and applied in the IPTW method will enhance readers' understanding of the robustness of our adjustments. We added the details and mechanisms on discussion section.

Comments 6: Mentioning potential limitations of IPTW, such as residual confounding, would provide a more balanced view.

Response 6: Thank you for your valuable suggestion. We agree that discussing the potential limitations of the IPTW method, including the issue of residual confounding, is essential for providing a balanced and transparent view of our study's findings. We addressed these limitations on method section.

Comment 7: More details on the Kaplan-Meier survival analysis, such as the censoring mechanism and the follow-up duration, would be beneficial.

Response 7: We appreciate the suggestion to provide more details on the Kaplan-Meier survival analysis. Including information on the censoring mechanism and follow-up duration would enhance the understanding of the study's methodology and results. The Kaplan-Meier method is a non-parametric statistic used to estimate the survival function from lifetime data, and it is crucial to describe how data censoring was handled and the duration of the follow-up period. We added the details on method section.

Comments 8: The discussion provides a good interpretation of the findings but could better differentiate between all-cause and liver-related mortality. It should more explicitly address why liver-related mortality was not significantly impacted post-IPTW.

Response 8: Thank you for your comments to better differentiate between all-cause and liver-related mortality in the discussion. We agree that a clearer distinction and an explicit explanation of why liver-related mortality was not significantly impacted post-IPTW would enhance the manuscript. We addressed the detailed contents on discussion section

Comments 9: There should be a clearer distinction between findings from this study and previous studies. Explicitly discussing why this study's results differ from past research would be beneficial.

Response 9: Thank you for your valuable comment. We agree that it is important to clearly distinguish between our findings and those of previous studies, as well as to discuss the potential reasons for any differences observed. We also added the contents on discussion section.

Comments 10: The proposed mechanisms for coffee's health benefits are well-explained but could be better linked to the study's findings. Consider discussing how the lack of impact on liver-related mortality fits with these mechanisms.

Response 10: Thank you for your insightful feedback. We agree that linking the proposed mechanisms for coffee's health benefits to our study's findings and discussing how the lack of impact on liver-related mortality fits with these mechanisms would provide a more comprehensive understanding of our data. We added the proposed mechanisms on discussion section.

Comments 11: The limitations section is brief. It should acknowledge other potential limitations, such as the observational design, potential measurement error in self-reported coffee consumption, and lack of differentiation between types of coffee.

Response 11: Thank you for your valuable feedback. We agree that acknowledging additional limitations, such as the observational design, potential measurement errors in self-reported coffee consumption, and the lack of differentiation between types of coffee, may provide a more comprehensive and balanced view of our study. We added the limitations in discussion section.

Comments 12: Discussing the generalizability of the findings to different populations or settings would be useful.

Response 12: Thank you for your suggestion. We agree that discussing the generalizability of our findings to different populations or settings is important for understanding the broader applicability of our results. We also added the contents in discussion section.

Comments 13: Consider adding a strengths section to highlight the robust aspects of the study, such as the large sample size and comprehensive adjustment for confounders.

Response 13: Thank you for your valuable comment. We appreciate the opportunity to highlight the strengths of our study, which include the large sample size and comprehensive adjustment for confounders, as these aspects significantly contribute to the reliability of our findings.

Comments 14: Comments on the Quality of English Language

Several typographical  errors are noted throughout the manuscript. for example: line 91 - "da", day.

Response 14: Thank you for pointing out the typographical errors in the manuscript. We appreciate your attention to detail. We conducted a review of the manuscript to correct all typographical errors and ensure the quality of the English language for the manuscript.

Reviewer 2 Report

Comments and Suggestions for Authors

 Interesting work on the relationships between a very widespread risk factor, exposure to coffee consumption, and general and specific mortality from liver disease on the UK Biobank cohort. This reviewer greatly appreciated the simple statistical analysis, using modern epidemiologic statistical techniques (IPTW) and the classic Kaplan Meyer, analyses permitted by the large size of the cohort.

These are our suggestions to possibly improve the work, at least in the discussion section:

1) English can be improved

2) The information in tables 1 to 4 is quite redundant. It is possible to eliminate the IPTW part of each table (to avoid making the work an exercise in epidemiological statistics). It is probably also possible to merge tables 2, 3 and 4 into a single table, leaving only the centrality indices of each variable (including coffee consumption stratified) by liver disease category at baseline (without steatosis, MASLD, MetALD) .

3) Unfortunately the exposure variable has definition problems: American coffee or espresso? arabica or robusta coffee? with caffeine or decaffeinated? if decaffeinated, with what method?  consumed alone or with the addition of milk? added teaspoons of sugar? if so, how many on average ? etc

4) There are also serious problems with controlling for confounding variables. It is not clear whether social class/income has been controlled. Coffee costs and consuming more cups of coffee can be associated with income/social class, in turn associated with general and specific mortality. Added sugar (or other sweeteners) associated with coffe  is also a confounding factor and does not seem that has been  controlled for .

Author Response

Comments 1: Interesting work on the relationships between a very widespread risk factor, exposure to coffee consumption, and general and specific mortality from liver disease on the UK Biobank cohort. This reviewer greatly appreciated the simple statistical analysis, using modern epidemiologic statistical techniques (IPTW) and the classic Kaplan Meyer, analyses permitted by the large size of the cohort.

These are our suggestions to possibly improve the work, at least in the discussion section:

Response 1: Thank you for your positive feedback on our work and for appreciating the statistical analysis methods we utilized. We are grateful for your insightful suggestions to further improve our manuscript, especially in the discussion section. Below, we address each of your suggestions in detail.

Comments 2: English can be improved

Response 2: Thank you for your feedback regarding the quality of the English language in our manuscript. We appreciate your suggestion and agree that improving the readability of the text is important. We reviewed whole manuscript for the grammar errors or typo.

Comments 3: The information in tables 1 to 4 is quite redundant. It is possible to eliminate the IPTW part of each table (to avoid making the work an exercise in epidemiological statistics). It is probably also possible to merge tables 2, 3 and 4 into a single table, leaving only the centrality indices of each variable (including coffee consumption stratified) by liver disease category at baseline (without steatosis, MASLD, MetALD) .

Response 3: Thank you very much for your thoughtful suggestions regarding the tables. We appreciate your concern about redundancy and the potential complexity added by including the IPTW parts in each table. However, we believe that maintaining the current format, including the IPTW-adjusted parts, is important for several reasons.
Including both pre- and post-IPTW data helps to transparently present the effect of the weighting process on balancing the covariates across groups. This detail is crucial for readers to assess the validity of our methodological approach.
Next, the standardized mean differences (SMD) provided in the IPTW parts of the tables are essential for demonstrating the balance achieved through the weighting process. These metrics are a standard way to show the effectiveness of IPTW in observational studies.
In addition, although merging tables could reduce the number of tables, it might also complicate the presentation by combining too much information in a single table. This could make it harder for readers to parse the key findings and understand the distribution of baseline characteristics across different liver disease categories.
Finally, separating the tables by liver disease category (no steatosis, MASLD, MetALD) allows for a clearer comparison and specific insights into each subgroup, which is a significant aspect of our study.
Given these considerations, we respectfully suggest retaining the current format of the tables to ensure a detailed and comprehensive presentation of our results. We hope this explanation addresses your concerns and helps to clarify the rationale behind our decision. Thank you again for your valuable feedback. We believe it has helped us improve the clarity and quality of our manuscript.

Comments 4: Unfortunately the exposure variable has definition problems: American coffee or espresso? arabica or robusta coffee? with caffeine or decaffeinated? if decaffeinated, with what method?  consumed alone or with the addition of milk? added teaspoons of sugar? if so, how many on average ? etc

Response 4: Thank you for your comment regarding the definition of the exposure variable. We acknowledge that the lack of detailed differentiation between types and preparation methods of coffee is a limitation in our study. We precisely added the contents on discussion section.

Comments 5: There are also serious problems with controlling for confounding variables. It is not clear whether social class/income has been controlled. Coffee costs and consuming more cups of coffee can be associated with income/social class, in turn associated with general and specific mortality. Added sugar (or other sweeteners) associated with coffe  is also a confounding factor and does not seem that has been  controlled for .

Response 5: Thank you for highlighting these important points regarding the control of confounding variables. We acknowledge the potential impact of social class/income and the addition of sugar or other sweeteners to coffee on our findings. We added the contents regarding to the limitations in discussion section.

Round 2

Reviewer 1 Report

Comments and Suggestions for Authors

I appreciate authors for meticulously addressing all the comments and revising the manuscript.

The revised manuscript reads impressive and I have no further comments to add.